# Declining Levels of Neutralizing Antibodies to SARS-CoV-2 Omicron Variants Are Enhanced by Hybrid Immunity and Original/Omicron Bivalent Vaccination

**DOI:** 10.3390/vaccines12060564

**Published:** 2024-05-22

**Authors:** Sharon Walmsley, Majid Nabipoor, Freda Qi, Leif Erik Lovblom, Rizani Ravindran, Karen Colwill, Roya Monica Dayam, Tulunay R. Tursun, Amanda Silva, Anne-Claude Gingras

**Affiliations:** 1Division of Infectious Diseases, University Health Network, Toronto, ON M5G 2C4, Canada; rizani.ravindran@uhn.ca; 2Department of Medicine, University of Toronto, Toronto, ON M5S 1A1, Canada; 3Biostatistics Department, University Health Network, Toronto, ON M5G 2C4, Canada; nabipoor@thebru.ca (M.N.); erik.lovblom@thebru.ca (L.E.L.); 4Lunenfeld-Tanenbaum Research Institute, Mount Sinai Hospital, Sinai Health, Toronto, ON M5G 1X5, Canada; fqi@lunenfeld.ca (F.Q.); colwill@lunenfeld.ca (K.C.); rdayam@lunenfeld.ca (R.M.D.); ttursun@lunenfeld.ca (T.R.T.); gingras@lunenfeld.ca (A.-C.G.); 5DATA Team, University Health Network, Toronto, ON M5G 2C4, Canada; amanda.silva@uhn.ca; 6Department of Molecular Genetics, University of Toronto, Toronto, ON M5S 1A1, Canada

**Keywords:** SARS-Cov-2, variant, vaccine, hybrid immunity, breakthrough infection, neutralizing antibody, Omicron

## Abstract

We determined neutralizing antibody levels to the ancestral Wuhan SARS-CoV-2 strain and three Omicron variants, namely BA.5, XBB.1.5, and EG.5, in a heavily vaccinated cohort of 178 adults 15–19 months after the initial vaccine series and prospectively after 4 months. Although all participants had detectable neutralizing antibodies to Wuhan, the proportion with detectable neutralizing antibodies to the Omicron variants was decreased, and the levels were lower. Individuals with hybrid immunity at the baseline visit and those receiving the Original/Omicron bivalent vaccine between the two sampling times demonstrated increased neutralizing antibodies to all strains. Both a higher baseline neutralizing antibody titer to Omicron BA.5 and hybrid immunity were associated with protection against a breakthrough SARS-CoV-2 infection during a 4-month period of follow up during the Omicron BA.5 wave. Neither were associated with protection from a breakthrough infection at 10 months follow up. Receipt of an Original/Omicron BA.4/5 vaccine was associated with protection from a breakthrough infection at both 4 and 10 months follow up. This work demonstrates neutralizing antibody escape with the emerging Omicron variants and supports the use of additional vaccine doses with components that match circulating SARS-CoV-2 variants. A threshold value for neutralizing antibodies for protection against reinfection cannot be determined.

## 1. Introduction

Despite the dramatic effectiveness of COVID-19 vaccines in preventing severe illness, hospitalization, and death due to COVID-19 infection [1,2], breakthrough infections are common, with 50–80% of most populations having acquired natural infection [3,4,5,6,7] after immunization. Diverse SARS-CoV-2 variants with genetic, antigenic, and phenotypic divergence from the preceding strains have caused successive waves of infection post-vaccine [8,9]. These variants of concern (VOC) often have higher transmissibility and increased ability to evade neutralizing antibodies acquired through infection or vaccination [10]. Vaccines with increased activity against the variant strains continue to be developed with the ultimate goal of a pan-Corona vaccine [11,12,13,14,15].

With vaccine fatigue and declining population vaccine uptake, correlates of protection from infection continue to be explored. We, among others, have shown that binding antibody levels cannot define protection or guide the frequency of vaccine boosters [16,17,18,19]. Although studies have suggested that neutralizing antibodies are more predictive, no threshold values have been established, and cross protection against variant strains has had limited study [20,21,22,23]. The development of potent neutralizing antibodies against SARS-CoV-2 appears to increase survival from infection [24].

Hybrid immunity resulting from three or more exposures to the virus antigen (i.e., one or more exposures from vaccination and one or more from SARS-CoV-2 infections before or after vaccination) may provide superior neutralization capacity against VOCs, including Omicron, compared with two doses of vaccination or previous SARS-CoV-2 infection without vaccination [23,25,26,27,28,29,30,31]. The waning of hybrid immunity, particularly due to Omicron infections, is not yet characterized in magnitude or duration, and it is unclear whether this protection will persist with new variants [26,32,33,34]. More data are needed for a precise quantification of the immune protection from hybrid immunity compared with vaccine-induced immunity normalized for the same antigen exposure. Both the quality and durability of hybrid immunity are likely to vary across age groups and in individuals with underlying medical conditions [35].

We set out to compare measures of serum-neutralizing antibodies to the original Wuhan SARS-CoV-2 strain and circulating VOC-Omicron BA.5, XBB.1.5 and EG.5 in the blood of participants who had evidence of both natural infection and vaccine (hybrid immunity) to those who received mRNA vaccines but have not demonstrated evidence of natural infection.

We hypothesized that those with hybrid immunity would have higher levels of neutralizing antibodies relative to those with vaccine immunity only. In addition, those who received the Pfizer-BioTech Comirnaty Omicron BA.4/BA.5 or the Moderna Spikevax bivalent BA.4/BA.5 vaccine booster would have higher levels of neutralizing antibody to this variant relative to those who did not and are less likely to have a breakthrough infection with this variant.

## 2. Materials and Methods

### 2.1. Study Design

The STOPCoV study has been described [36] and is ongoing. The full protocol is available on the study website (www.STOPCov.ca, accessed on 10 April 2024). In short, this decentralized study enrolled 1286 participants across Ontario, Canada, beginning in 2021. We enrolled two cohorts, namely those aged >70 years and those aged 30–50 years, with the goal of comparing antibody responses to COVID-19 vaccines. We prospectively collect electronic self-reported data on COVID-19 vaccine brands, doses, and breakthrough infections and determine binding IgG antibody levels (BAU/mL-binding antibody units per milliliter) to the spike trimer (S), its receptor-binding domain (RBD), and nucleoprotein (NP) at 3 monthly intervals and after booster vaccine doses through enzyme-linked immunosorbent assays (ELISA) on self-collected dried blood spots (DBS) [37]. As vaccines approved in Canada only contain the spike antigen, positivity to a nucleoscapsid is an indication of past infection.

### 2.2. Recruitment to the Sub-Study

In the fall of 2022, a random selection of STOPCoV participants who were identified as having either hybrid or vaccine-only immunity and who consented to be contacted for future research were email-invited to participate in this sub-study. Participants with one or more documented breakthrough infections before the baseline visit of this sub-study were categorized as having hybrid immunity, while those without any breakthrough infections were categorized as having vaccine-only immunity. All participants had received the initial two-dose vaccine series between 26 May and 16 September 2021.

Given that we over-recruited the older cohort in the main study, we included 126 in the older (≥70 years of age) and 52 in the younger cohort (30–50 years) in this sub-study to maintain the age distribution. Half of the participants in each cohort had vaccine only and half had hybrid immunity at the baseline testing of this sub-study. COVID-19 infection was determined by either a report of a positive PCR or RAT (rapid antigen test) to COVID-19 or the development of nucleocapsid antibodies passing the threshold of seropositivity [36].

### 2.3. Participant Consent

Consent was completed electronically as in the main study with interspersed questions and correct answers provided to ensure comprehension. Study-staff contact information (email and telephone) was provided for assistance. This sub-study was approved by the University Health Network (UHN) research ethics board (Study 22-5715).

### 2.4. Sample Collection

Upon consent, blood collection kits were sent by the study team to the enrolled participants. An external lab collected and packaged specimen samples in barcoded containers sent to the lab at the UHN for processing. The collection and shipping of specimens were performed at baseline (December 2022) and 4 months later (April 2023) to assess the longevity of neutralizing antibodies in those with and without hybrid immunity. Breakthrough COVID-19 infections were documented until September 2023.

### 2.5. Laboratory Analysis

Separated serum was transferred to the Lunenfeld–Tanenbaum Research Institute (LTRI) at Sinai Health for processing (Sinai REB study 23-0069-E).

#### 2.5.1. Spike-Pseudotyped Lentivirus Neutralization Assays

The lentivirus neutralization assays were performed to assess the neutralization capacity against wildtype (Wuhan D614G) SARS-CoV-2 and Omicron variants of concern (VOC) BA.5, XBB.1.5. and EG.5. Spike cDNA variants were synthesized at Twist Bioscience (San Francisco, CA, USA) using https://outbreak.info/ and https://viralzone.expasy.org/9556 (accessed on 11 August 2023) to identify the amino acid changes. The cDNAs were cloned into the EcoRI and XhoI sites of HDM_IDTSpike_fixK (gift from Jesse Bloom (Fred Hutchinson Cancer Center, Seattle, WA, USA); available at BEI Resources catalog NR-52514; plasmid maps and sequences for the VOCs are available at nbcc.lunenfeld.ca/resources/ (accessed on 11 August 2023)). The generation of spike-pseudotyped lentivirus particles was performed as described previously with minor modifications [38]. Briefly, HEK293TN cells (LV900A-1, System Biosciences, Palo Alto, CA, USA) were transiently co-transfected with 0.4 μg HDM Wuhun-1 D614G or the Omicron sub-variants BA.5, XBB.1.5, and EG. 5 (along with packaging plasmid (1.3 μg psPAX2, catalog 12260, Addgene, Watertown, MA, USA) and reporter constructs (1.3 μg luciferase-expressing pHAGE-CMV-Luc2-IRES-ZsGreen-W provided by Jesse Bloom and Katharine Crawford, Fred Hutchison Cancer Research Center, Seattle, WA, USA) using 7 μL of Lipofectamine 3000 Transfection Reagent and 6 μL P3000 Enhancer Reagent (Thermo Fisher Scientific, catalog L3000015) per well of a 6-well plate containing 2 mL of growth medium (10% FBS, 1% penicillin/streptomycin). After 6 h of transfection, the medium was replaced by 3 mL of DMEM (Dulbecco’s Modified Eagles’s Medium) containing 5% FBS (fetal bovine serum) and 1% penicillin/streptomycin, and cells were incubated for 16 h at 37 °C and 5% CO_2_ before transferring them to 33 °C and 5% CO_2_. Approximately 52–54 h post-transfection, the supernatant was collected, filtered through a 0.45 μm filter, and frozen at −80 °C.

For the neutralization assay, the serum samples were heat-inactivated for 30 min at 56 °C, serially diluted, and incubated with the lentiviral particles for 1 h at 37 °C prior to addition to cells (HEK293T-ACE2/TMPRSS2), as previously described [38]. After 48 h, luminescence signals were detected with the Bright-Glo Luciferase assay system (catalog E2620, Promega, Madison, WI, USA) on an EnVision multimode plate reader (Perkin Elmer, Waltham, MA, USA). This assay, using pseudotyped Wuhun-1 D614G spike lentivirus, has been calibrated to the first WHO International Standard (National Institute for Biological Standards and Control (NIBSC) South Mimms, UK, Code 20/136) and validated using the first WHO International Reference Panel (Code 20/268) [39].

#### 2.5.2. Half Maximal Inhibitory Dilution/Concentration (ID50 or IC50) Calculation

Relative luciferase units were first normalized to positive and negative controls. A sigmoidal curve with the association Y = 100/(1 + 10^((Log10ID50 − x) × HillSlope))) was fitted, and an ID50 value was calculated for each sample, using the nlsLM function from the minpack.lm package in R [40,41]. Samples that did not show a neutralization capacity at the serum dilutions performed were assigned an ID50 value of 1.

### 2.6. Statistical Analysis

Participant characteristics, according to hybrid or vaccine-only immunity at the baseline visit, were compared using chi-squared tests, Fisher’s exact tests, or Student’s *t*-tests as appropriate. The demographics of the sub-study cohort were compared to the STOPCoV cohort as a whole using a family-wise Bonferroni correction test.

Neutralizing antibody titers at baseline to the Wuhan strains and the other variants were analyzed at two levels: (1) detection above the threshold and (2) among the detectable, the neutralizing antibody levels (log10 (1/ID50)). The association between the detection of antibody levels at the baseline visit and the predictor variables, such as age group, gender, obesity, comorbidity, hybrid/vaccine-only immunity, number of vaccine doses to the baseline visit, and infection/vaccination time to the baseline visit, was examined using logistic regression. The association between the detectable antibody levels at the baseline visit and predictor variables was investigated using linear regression on the log-transformed values.

We examined the trajectories of neutralizing antibody levels between the baseline and 4-month follow-up sampling times. As all participants displayed detectable neutralizing antibodies to Wuhan at both time points, two trajectory patterns were observed: decreasing and increasing. For the other variants, as not all participants had detectable neutralizing antibodies at baseline, four trajectory patterns were identified: positive–zero, decreasing, increasing, and zero–positive. The slopes of the regression lines representing these trajectory patterns from baseline to the 4-month follow up were determined. The association between the sorted patterns based on slopes of trajectories and predictor variables was examined using logistic regression for Wuhan and an ordinal logistic regression model for the VOC Omicron BA.5, XBB.1.5, and EG.5. We conducted a descriptive analysis for those who did not receive an additional vaccine dose or experienced a breakthrough infection to examine the decay slopes of the neutralizing antibody levels.

To examine protective factors against a new breakthrough infection in the short-term (4 months) and longer-term period (10 months) after the baseline visit, up to 22 September 2023 (when the XBB monovalent vaccine became available in Ontario), a Cox regression analysis was conducted. The time of the first breakthrough infection from the baseline visit date up to the 4-month follow up and to 22 September 2023 was used as the outcome for the short-term/longer-term analysis. The analysis included fixed time-independent predictor variables, such as age groups, gender, obesity, comorbidity, hybrid/vaccine-only immunity at baseline, vaccination during the short term/long term, and breakthrough infection during the respective period. As the predictors of hybrid/vaccine-only immunity and neutralizing antibody titers to BA.5 at baseline are interdependent, two distinct multivariate Cox regressions were performed to evaluate their respective effects.

Univariable and multivariable models were constructed. In linear regression, the coefficient (β, adβ) represents the effect of the predictor in the model, indicating whether it has an increasing/decreasing effect. The reference level for the coefficients (β, adβ) is zero, unlike the ratios, such as odds ratios (OR, adOR) in logistic regression and hazard ratios (HR, adHR) in survival analysis, where the reference level is one. Tests of proportional odds/hazards were conducted. A significance level of α = 0.05 was used for all statistical hypothesis tests. Data manipulation and analyses were performed using SAS (version 9.4) and R software (version 4.2.2).

## 3. Results

We enrolled 178 participants, of whom 92 had hybrid immunity (65 in the older cohort and 27 in the younger cohort), and 86 had vaccine-only immunity (61 in the older and 25 in the younger cohort). The study flow chart is shown in Figure 1. The demographic characteristics and vaccination status of the participants are shown in Table 1.

Similar proportions of the hybrid and vaccine-only cohorts were women, Caucasians, and non-smokers. Approximately half of each cohort had reported at least one underlying comorbidity. The sub-study participants were representative of those of the main STOPCoV study based on the Boforreni family-wise error threshold of 0.005, Appendix A.

The hybrid group had received a median of four, while the vaccine-only group had a median of five vaccine doses at baseline (*p* < 0.009). The difference likely reflects the recommendations for deferral of a vaccine booster by 6 months after a natural infection. Between the baseline visit and month 4 follow up, 26%/30% of each group (hybrid immunity and vaccine only) received another vaccine dose (Pfizer-BioNTech Comirnaty Omicron BA.4/BA.5 or Moderna Spikevax bivalent BA.4/BA.5). Twenty-four of the vaccine-only group and eight of the hybrid-immunity group had a breakthrough COVID-19 infection within the 4 months follow up between sampling times, during the Omicron BA.5 wave in Ontario.

The results of the neutralizing antibody tests at baseline and the 4-month follow up of the sub-study for the Wuhan ancestral strain and the Omicron BA.5, XBB.1.5, and EG.5 variant strains are illustrated in Figure 2. Participants with hybrid immunity at baseline exhibited higher antibody levels across the four strains and were less likely to have non-detectable neutralizing antibody levels to Omicron BA.5, XBB.1.5, and EG.5 at baseline and in follow up.

A neutralizing antibody response to the Wuhan ancestral strain at baseline was detected in all participants, with a significantly higher mean antibody of 4.28 log10 (1/ID50) for the hybrid cohort compared to 3.79 log10 (1/ID50) for the vaccine-only cohort (*p* = 0.0001), as shown in Table 2.

Overall, at the baseline visit, the proportion of participants with detectable neutralizing antibodies to each of the variants was decreased relative to Wuhan. Neutralizing antibodies to Omicron BA.5 were found in 92.1% of participants, while this was reported in 65.2% to Omicron XBB1.5 and 56.7% to Omicron EG.5. While 83–90% of those with hybrid immunity had a detectable neutralizing antibody to Omicron XBB.1.5 and Omicron EG.5 variants, this was only detectable in 29–40% of those with vaccine-only immunity. For those with detectable neutralizing antibodies the levels decreased with the evolution of the variants but, in each case, were higher for those with hybrid immunity, Figure 3.

The average antibody level at baseline for the Omicron BA.5, XBB.1.5, and EG.5 variants was 3.64–2.62 log10 (1/ID50) for those with hybrid immunity, whereas it was only 2.62–2.04 log10 (1/ID50) for those with vaccine-only immunity. The trends in detectable neutralizing antibodies and the average levels for the variants showed the same pattern in the two age cohorts (≥70 years, 30–50 years), as shown in Appendix A.

The detection rates of neutralizing antibodies at baseline to the Omicron BA.5 (92%), Omicron XBB.1.5 (65%), and Omicron EG.5 (57%) variants were considerably lower than that of Wuhan (100%). We explored predictors of detectable neutralizing antibodies, namely age group, gender, obesity, comorbidity, hybrid/vaccine-only immunity at baseline, number of vaccine doses up to the baseline visit date, and infection/vaccination time to baseline. Our logistic univariable/multivariable analysis (Appendix A, Figure 4 (upper panel)) revealed that only hybrid immunity had a significant association with detectable neutralizing antibodies to Omicron BA.5 with an adjusted (ad) OR (odds ratio) of 12.67 (95% CI 2.33, 68.84), Omicron XBB 1.5, with an adOR of 17.86 (95% CI 7.29, 43.76), and Omicron EG.5 with an adOR of 13.36 (95% CI 6.13, 29.11).

Of those with detectable neutralizing antibodies, higher levels at baseline were found in those with hybrid immunity relative to those with vaccine-only immunity (*p* < 0.0001), as shown in Figure 4 (upper panel) and Appendix A. We investigated the predictors of age cohort, gender, obesity, comorbidity, and infection/vaccination time to baseline on the level of neutralizing antibodies at baseline by multivariate analysis using linear regression. The presence of hybrid immunity had a significantly increasing effect on the baseline neutralizing antibody levels to Wuhan; with an adjusted (ad) beta of 0.63 (95% CI 0.38, 0.88), to Omicron BA.5; with an adB of 1.08 (95% CI 0.89, 1.27), to Omicron XBB.1.5; with an adB of 1.08 (95% CI 0.89, 1.27, and Omicron EG.5; with an adB of 0.61 (95% CI 0.38, 0.83). The number of vaccine doses up to the baseline sample had a significant increasing effect, with an adB of 0.21 (95% CI 0.01, 0.41) for Wuhan, but not for the variants, as shown in Appendix A and Figure 4 (lower panel).

### Baseline to 4-Month Follow-Up Analysis

We then evaluated the change in neutralizing antibody levels between the baseline and the 4-month follow-up visit. The proportion of participants without detectable neutralizing antibodies either at baseline or the 4-month follow up increased with the evolution of variants from Wuhan (0%), Omicron BA.5 (14.6%), and Omicron XBB.1.5 (44.9%), to Omicron EG.5 (51.7%), as shown in Appendix A.

We identified two patterns of response for those with detectable antibodies to Wuhan at baseline—either increasing or decreasing. For the other variants, four patterns were observed, positive to zero, decreasing, increasing, and zero to positive, Figure 5, Appendix A.

We sorted the identified pattern groups in ascending order based on the observed significant slopes, ranging from the positive–zero group to the decreasing group, followed by the increasing group, and finally, the zero–positive group to enable illustration of the changes in neutralizing antibody titers during the 4-month follow-up period. The association between the sorted slopes of trajectories and predictor variables, including age group, gender, obesity, comorbidity, hybrid/vaccine-only immunity at baseline, vaccination during the 4-month follow up, and breakthrough infection during the same period, was examined using logistic regression for the Wuhan strain and an ordinal logistic regression model for the Omicron BA.5, XBB.1.5, and EG.5 variants. We explored the predictors of these changes (Table 3, Appendix A) and observed that vaccination during the 4-month follow-up period significantly increases the likelihood for a higher neutralizing antibody level at the 4-month follow up for all variants: Wuhan; with an adjusted odds ratio (adOR) 5.79 (95% CI 2.64, 12.72), Omicron BA.5; with an adOR 7.08 (95% CI 3.39, 15.28), Omicron XBB.1.5; with an adOR 5.58 (95% CI 2.78, 11.58), and Omicron EG.5 with an adOR 4.43 (95% CI 2.26, 8.91). Additionally, a breakthrough infection during the 4-month follow-up period significantly increased the likelihood for higher neutralizing antibody levels at the 4-months follow up, and this increasing effect is larger compared to vaccination during the follow-up period for Omicron BA.5; with an adOR 20.11 (95% CI 7.94, 55.33), Omicron XBB.1.5; with an adOR 77.94 (95% CI 28.04, 236.54), and Omicron EG.5; with an adOR 73.96 (95% CI 26.66, 224.96), Table 3.

Of note, there was a regression to the mean with time (Appendix A). The average antibody levels at baseline are higher for the group that demonstrated decreasing levels over the follow up, while the average antibody levels at the 4-month follow up are higher for the group that showed increased antibody levels with time. Consequently, the mean-value lines for the increased and decreased groups crossed for all four variants, as depicted in Appendix A.

For those who did not have either a new vaccine or a breakthrough infection between the baseline and 4-month sampling times, we were able to determine the slope of decay of the neutralizing antibodies as shown in Figure 6 and Table 4.

We observed that the hybrid-immunity group always had higher neutralizing antibody levels to Wuhan and all Omicron variants compared to the vaccine-only group, and in both groups, there was decay over the 4-month follow-up period. The decay was significant among individuals in the vaccine-only group to Wuhan and all Omicron variants but was not statistically significant in the hybrid-immunity group.

We then explored protective factors against breakthrough infection (Appendix A). During the short-term (4-month) follow-up period, there were 32 breakthrough infections observed, with 8 (25.0%) occurring in participants with hybrid immunity at baseline, and 24 (75.0%) occurring in those with vaccine-only immunity (*p* = 0.0009). During the longer-term (10-month) follow-up period from the baseline visit to 22 September 2023, there were 68 breakthrough infections observed, with 31 (45.6%) occurring in participants with hybrid immunity at baseline and 37 (54.4%) occurring in those with vaccine-only immunity (*p* = 0.2006). An additional vaccine dose was received by 15.6% of participants before experiencing the breakthrough infection in the short-term period; this percentage rose to 26.5% during the longer-term period.

We investigated the predictors of age, gender, obesity, comorbidity, hybrid immunity at baseline, and vaccination during the period before breakthrough infection (Table 5, Figure 7).

Given that the presence of baseline hybrid immunity and neutralizing antibody levels to Omicron BA.5 were correlated, we ran two separate models. That on the left-hand side includes hybrid-immunity status and that on the right includes the baseline neutralizing antibody titer. We observed that hybrid immunity at baseline was significantly associated with protection against a breakthrough infection in the short term, with an adHR of 0.24 (95% CI 0.11, 0.54), but not in the longer term. Similarly, in the separate model, higher neutralizing antibody levels to Omicron BA.5 were also associated with a lower rate of breakthrough infection in the short term, with an ad HR of 0.72 (95% C.I. 0.55, 0.94), but not in the longer-term follow up during the time when this was the main circulating variant. The most significant protective association against breakthrough infection was the receipt of an additional vaccination (Original/Omicron BA.4/5 bivalent) in the short term, with an adHR of 0.35 (95% CI 0.13, 0.93), and the longer-term period, with an adHR of 0.4 (95% CI 0.23, 0.7).

## 4. Discussion

We examined the neutralizing capacity of serum from 178 adults in a highly vaccinated cohort to the original Wuhan SARS-CoV-2 strain and several of the Omicron variants 15–19 months after the initial two-dose mRNA COVID-19 vaccine series. At the time of our sub-study, the participants had received four to five vaccine doses, primarily mRNA1273 and BNT162b2 [36]. We demonstrated that, although all the participants had evidence of neutralizing antibodies to the original Wuhan strain, the proportion with detectable antibodies to the Omicron variants BA.5, XBB.1.5, and EG.5 was considerably lower, suggesting an escape of the variants from neutralization. Further, for those with detectable antibodies, the level of neutralizing antibodies to the variants was also lower, by 1.5 to 2 log 10 1/ID 50 relative to that of Wuhan. Our data are similar but demonstrate a lesser decline in cross-variant neutralizing antibody levels than that demonstrated in other work. In a study of five clinical cohorts of individuals who received three to four doses of the original COVID-19 mRNA vaccines, Wang et al. [10] showed significant evasion of serum neutralization for Omicron BA.2 and BA.4/5 variants with 3- to 14-fold decreases in neutralizing titers and with reductions of >71 fold to Omicron XBB.1.5. In their work, the subvariants were barely susceptible to neutralization, even after boosting with Original/Omicron BA.4/5. However, it is unclear at the amount of time between vaccination and testing. Likewise, Reinholm et al. [42] showed that three-dose vaccination combinations induced significant levels of neutralizing antibodies against older SARS-Cov-2 variants in 432 Finish healthcare workers but were less effective at inducing antibodies against the BQ.1 and XBB.1.5 variants. Although most had detectable levels of neutralizing antibodies to Omicron BQ1.1 and XBB.1.5 immediately after the third vaccine dose, the number declined rapidly with time.

It is unclear whether more doses of the original vaccine result in a better neutralizing capacity against the Omicron variants. It has been reported that vaccination with mRNA vaccines can enhance neutralizing activities across a broad range of variants, and booster doses of homologous mRNA vaccines have been shown to increase the neutralizing antibody response against Omicron variants [32]. Perez et al. [43] were unable to identify the neutralization of Omicron in participants with two doses of CoronaVac, but a BNT 162b2 booster resulted in a 1.4-fold increase in neutralizing activity to Omicron compared with the two-dose mRNA vaccine. However, neutralizing antibody titers were reduced by 7.1 fold and 3.6 fold for Omicron compared to the ancestral strain and the Delta variant. Our group has previously shown that an Omicron breakthrough infection significantly increased the number of interferon alpha-secreting cells compared to individuals who had been vaccinated twice [44]. Both breakthrough infections and third vaccine doses potentiated the broadness and the magnitude of the T-cell responses. In the current study, the multivariable analysis revealed that the number of vaccine doses did result in a greater proportion with detectable neutralizing antibodies to Omicron BA.5 but not Omicron XBB.1.5 nor Omicron EG.5. For those with detectable neutralizing antibodies, a greater number of vaccine doses resulted in higher neutralizing antibody levels to Wuhan but did not result in higher antibody levels to any of the other variants evaluated, thus adding to the concern for escape.

Our data are consistent with other studies that show less neutralizing antibodies following vaccination with the ancestral spike-based vaccines to the emerging variants [28,29,30,42,43]. This raised concern about vaccine efficacy and led to the development of bivalent vaccines that included ancestral spike and the Omicron BA.5 spike proteins [34]. We have shown that the receipt of the Original/Omicron BA.4/5 vaccine resulted in increased neutralizing antibody levels not only to the Wuhan strain but also to the Omicron BA.5, XBB1.5, and EG.5 variants. Receipt of this additional vaccine also resulted in fewer breakthrough infections in 4–10 months of follow up. Similarly, Davis-Gardner et al. [34] showed that persons who received either one or two monovalent COVID-19 vaccine boosters had lower neutralization titers against BA.2 and XBB variants. Likewise, Miller et al. [45] assessed 15 participants who received a monovalent mRNA booster and 18 participants who had received a bivalent booster in 2022 and demonstrated that the BQ1.1 and XBB.1 variants escaped neutralizing antibodies more effectively than the BA.5 variants in both groups.

We, like others, have shown that the presence of hybrid immunity is associated with higher proportions who develop detectable neutralizing antibodies and higher levels of neutralizing antibodies to both the Wuhan and Omicron variants [23,28,46]. In our study, although over 80% with hybrid immunity had detectable neutralizing antibodies to the variants at baseline, this was detected in 40% or less in the vaccine-only group. With each variant we studied, the participants who had hybrid immunity were more likely to have detectable antibodies and higher levels of neutralizing antibodies to all variants. This did not vary by age cohort (≥70 years vs. 30–50 years) or by race, the presence of underlying comorbidity, or gender. Again, those with hybrid immunity at the baseline of the study were less likely to have a breakthrough infection after 4 months but not at 10 months of follow up.

Given the poor outcomes of older adults to COVID-19 infection [47,48], concern had been raised as to the lower overall antibody responses and waning immunity in older vaccinees [49,50,51]. In the main STOPCoV study, we found that binding antibody responses were not different in our two age groups (>70 years and 30–50 years) [19,36]. In this sub-study, age group was not an independent predictor for lower neutralizing antibody responses to Wuhan or the variants after controlling for the number of vaccine doses. Whether the neutralizing responses decay more rapidly requires further study. Sanchez-Sendra et al. [52], in a study of sera from 30 nursing-home residents, demonstrated neutralizing antibody to VOC (Beta, Gamma, Delta, and Epsilon) one month after the second vaccine dose but with lower titers especially with the Beta variant relative to the Wuhan. The reduction in neutralizing antibody activity to the VOC was not significant from that of 18 healthy younger controls. They concluded that age, frailty, and comorbidity did not impact the serum-neutralizing activity against VOC. Newman et al. studied [35] 37 individuals aged 70–89 years and found that, between 3 and 20 weeks after the initial two-dose vaccine series, the neutralizing antibody titers fell 4.9 fold, with 22% not having neutralizing antibodies at the second time point. They did not have a comparator younger cohort. In contrast, all our participants had neutralizing antibodies to Wuhan 15 to 19 months after vaccination.

We also evaluated the change in neutralizing antibody detection and neutralizing antibody levels over a 4-month period. As anticipated, receipt of an additional vaccine dose (in this case the Original/Omicron BA.4/5 bivalent vaccine) was associated with increased neutralizing antibody levels to all variants but most strongly to the BA.5 variant. Over this period, developing a breakthrough infection (Omicron BA.5 wave) was associated with increased neutralizing antibodies to the Omicron BA.5, XBB.1.5, and EG.5 variants. This might suggest that this immune background may be sufficient to shape the immune response against the wave of the XBB.1.5 variant.

For those who did not experience a breakthrough infection or receive an additional vaccine during the 4-month follow up, we found that those with hybrid immunity had higher neutralizing antibodies at both time points to all strains, and decay of the neutralizing antibodies occurred in both groups but was only statistically significant in the group with vaccine-only immunity. This is consistent with the findings of improved protection in the group with hybrid immunity from subsequent breakthrough infection. However, the association with protection from breakthrough diminished with further follow up.

Higher neutralizing antibody titers to Omicron BA.5 were associated with protection from a breakthrough infection during the short-term follow-up. However, we were unable to define a threshold value for protection. Neither hybrid immunity nor the baseline neutralizing antibody titers were independently associated with protection from breakthrough infection in the longer-term follow-up, suggesting either the loss of neutralizing antibodyies with time or viral escape with new variants.

Similarly, Alonso et al. [20], in a study of healthy vaccinated healthcare workers, concluded that neither neutralizing antibody levels nor binding antibody levels measured at 30 days post-vaccination can be used as predictors of breakthrough infection. In the main STOPCoV study, no binding antibody levels for protection were identified [19]. Our study receipt of the Original/Omicron BA.4/5 vaccine showed the strongest association with protection from a breakthrough infection.

### Study Strengths and Limitations

Our study has many strengths. Our main study, from which this sub-study was recruited, was completely decentralized. Sera collected from an outside lab for the neutralization studies allowed for the inclusion of participants with limited mobility or distance from research facilities and included a high proportion of older persons who are at higher risk of poor outcomes from COVID-19 infection. The population studied had high vaccine and booster uptake and were followed longitudinally through multiple waves of the pandemic. We measured neutralizing antibodies 15–19 months after the initial vaccine series and 4 months later. We relied on participant reporting of infection and vaccine brands and booster doses, which could lead to some misclassification. For cases that were minimally or asymptomatic and no RAT or PCR was performed, the true date of infection was unclear, as antibody to nucleocapsid was only determined every three months. Lastly, the chosen nucleocapsid threshold of 99% specificity as an indication of infection may have misclassified cases with low antibody levels, if antibodies waned between testing periods, or if Paxlovid had been used. Lower antibody levels have been reported in a vaccinated population [53]. Receipt of a vaccine or breakthrough infection in the short period before our baseline studies may have impacted the trajectories of the neutralizing antibody responses.

## 5. Conclusions

The original mRNA vaccines elicited a strong neutralizing antibody response to Wuhan SARS-CoV-2, but responses were lower against the Omicron BA.5, XBB1.5, and EG.5 variants in a highly mRNA-vaccinated older population 15–19 months after the initial two-dose vaccine series. Only 30% of the vaccine-only group had detectable neutralizing antibodies to Omicron EG.5. Additional doses of the original COVID-19 vaccine increased neutralizing antibodies to Wuhan but not to the Omicron-variant strains. The bivalent Original/Omicron BA.4/5 vaccine increased the neutralizing antibody levels to Omicron BA.5 and XBB.1.5 and EG.5. Those with hybrid immunity have increased the neutralizing antibody levels to the Omicron variants, and this trend is consistent in both of our age cohorts, regardless of gender and comorbidity. The rate of decay of the neutralizing antibody levels to all variants over 4 months was significant in the vaccine-immunity-only group. In contrast to other studies, all of our highly vaccinated participants showed detectable neutralizing antibodies to Wuhan. Increased protection against breakthrough infection was observed in those who received an Original/Omicrons BA.4/5 bivalent vaccines. Antibody titers to Omicron BA.5 were associated with protection from breakthrough infection in short-term follow up during the Omicron BA.5 wave but not in the longer 10-month follow up. No threshold value for neutralizing antibody levels for protection against breakthrough infection with Omicron variants could be identified. Our data support the continued use of booster vaccines that match the circulating COVID-19 variants.

## Figures and Tables

**Figure 1 vaccines-12-00564-f001:**
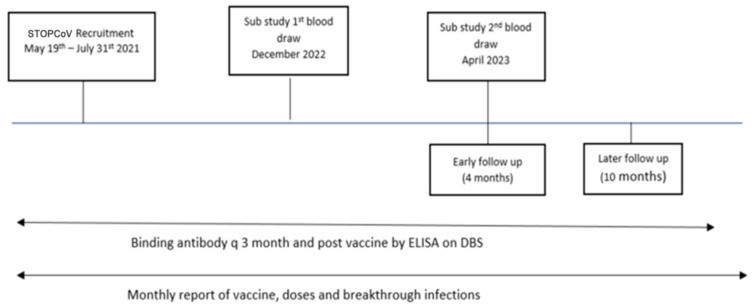
Timeline for main STOPCoV study and sampling times for neutralizing antibody sub-study. DBS—dried blood spot, ELISA—enzyme-linked immunosorbent assay.

**Figure 2 vaccines-12-00564-f002:**
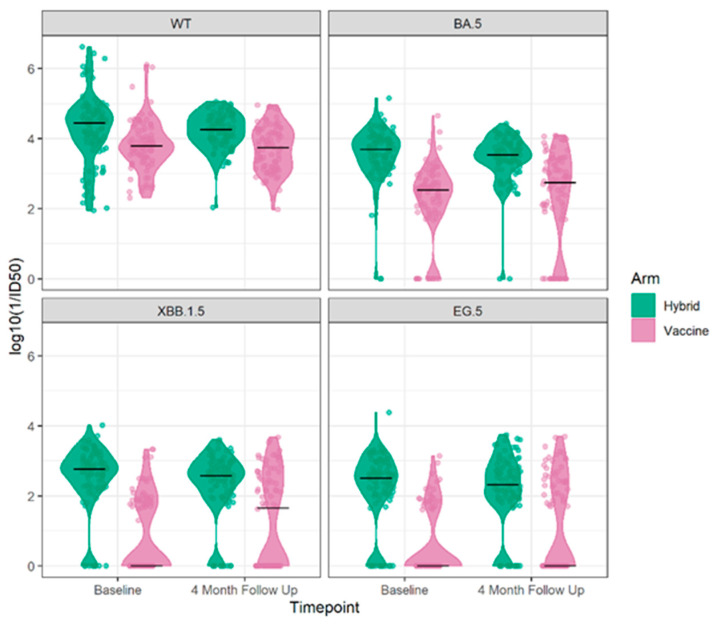
The neutralizing antibody results log 10 (1/ID 50) at the baseline and 4-month follow up for the Wuhan ancestral strain (**top left**) and to the Omicron BA.5 9 (**top right**), XBB.1.5 (**bottom left**), and EG.5 (**bottom right**) variant strains by cohort vaccine only or hybrid immunity.

**Figure 3 vaccines-12-00564-f003:**
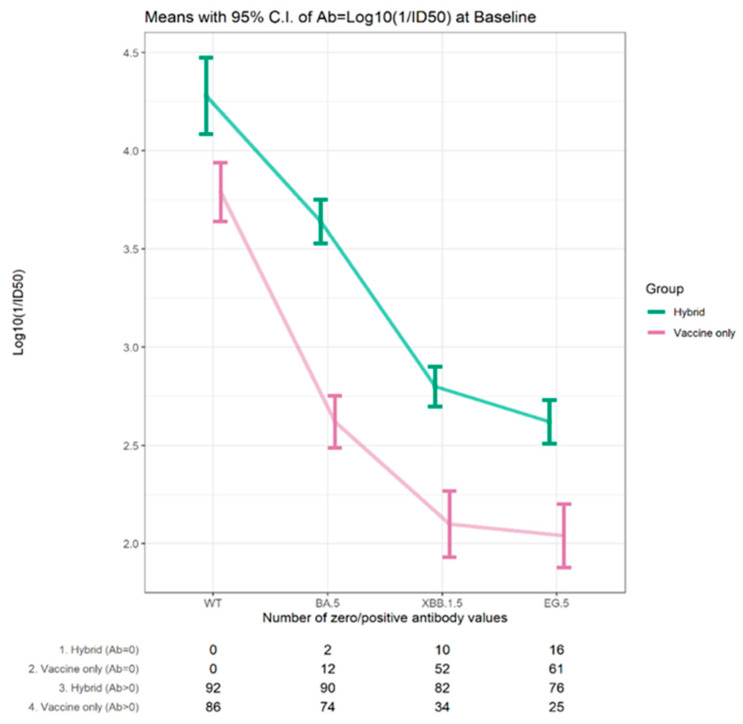
Mean neutralizing antibody titers (log 10 1/ID50) for those with detectable levels for Wuhan and Omicron variants at baseline for participants with hybrid/vaccine immunity at baseline.

**Figure 4 vaccines-12-00564-f004:**
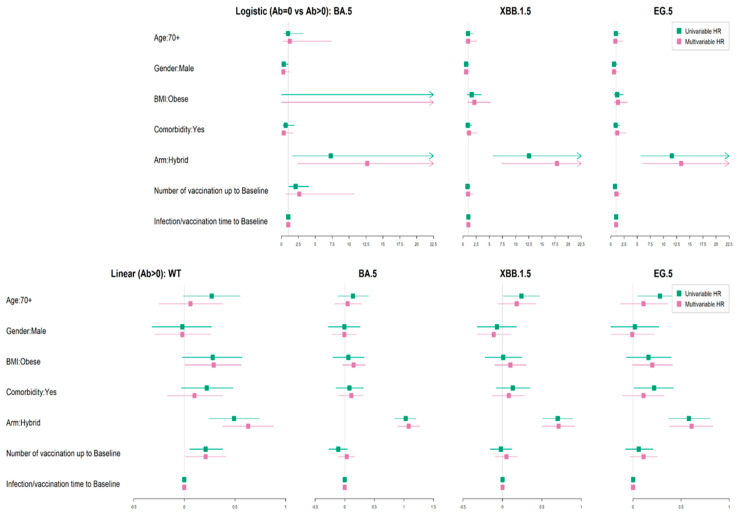
(**upper panel**) Logistic regression of factors influencing the detectability of the neutralizing antibody to Omicron BA.5, XBB.1.5, and EG.5 variants. (top 3 panels). (**lower panel**) Linear regression of factors influencing the neutralizing antibody titers to Wuhan and Omicron BA.5, XBB.1.5, and EG.5 variants in those with detectable levels. (bottom 4 panels).

**Figure 5 vaccines-12-00564-f005:**
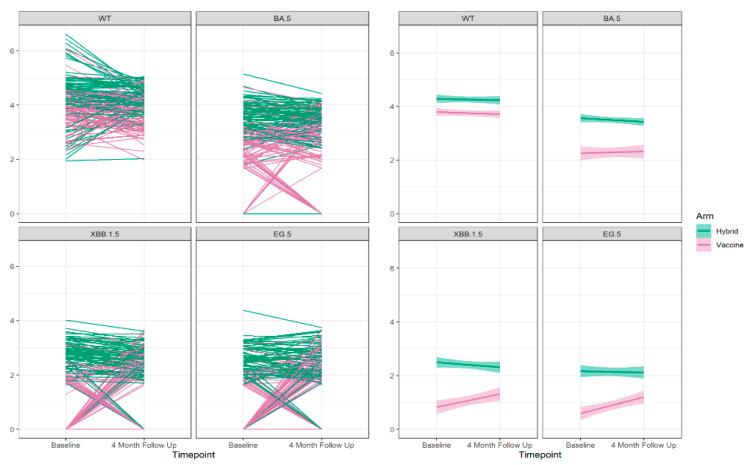
The neutralizing antibody trajectories (**left 4 panels**) and the overall regression lines (**right 4 panels**) for the Wuhan ancestral strain (WT) and the Omicron BA.5, XBB 1.5, and EG.5 variant strains during 4-months follow up.

**Figure 6 vaccines-12-00564-f006:**
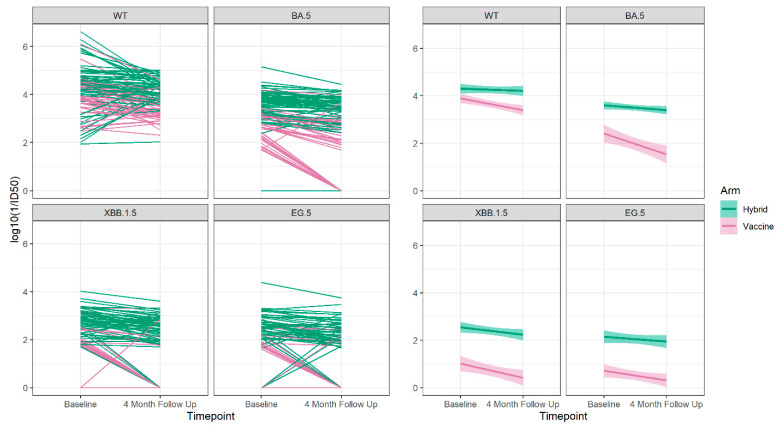
The neutralizing antibody trajectories (**left 4 panels**) and the overall regression lines (**right 4 panels**) for the Wuhan ancestral strain (WT) and the Omicron BA.5, XBB 1.5 and EG.5 variant strains during 4-months follow up for those who did not have a breakthrough infection or receive an additional vaccine dose between the baseline and 4-month sampling period.

**Figure 7 vaccines-12-00564-f007:**
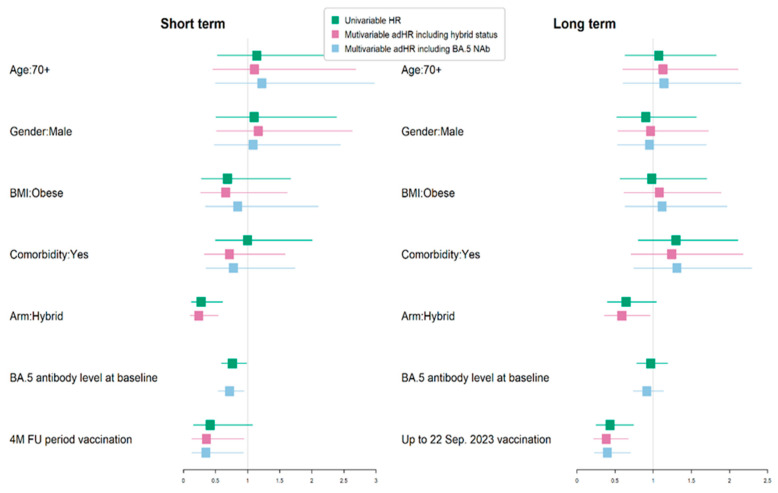
Univariate and multivariate analysis of protective factors against breakthrough COVID-19 infection in short (4-month)/long term (10-month) follow up (FU). BMI—body-mass index.

**Table 1 vaccines-12-00564-t001:** The demographic characteristics of vaccination history and COVID-19 infection status of the sub-study cohort. BMI—body-mass index, SD—standard deviation, IQR—interquartile range, n—number.

Characteristics	Hybrid Immunity	Vaccine-Only Immunity	Total	*p*-Value
**Total (N)**	92	86	178 (100%)	-
**Age, n (%)**				
30–50 years	27 (29.3%)	25 (29.1%)	52 (29.2%)	0.9675
70+ years	65 (70.7%)	61 (70.9%)	126 (70.8%)	
Mean (SD)	66.28 (15.67)	66.29 (15.03)	66.28 (15.32)	0.9956
Median (IQR)	73.6 (49.1, 76.8)	73.1 (50.7, 75.7)	73.3 (50.3, 76.3)	0.6321
**Gender, n (%)**				
Female	69 (75.0%)	61 (70.9%)	130 (73.0%)	0.5409
Male	23 (25.0%)	25 (29.1%)	48 (27.0%)	
**BMI, n (%)**				
Obesity (BMI ≥ 30)				
No	70 (76.1%)	63 (73.3%)	133 (74.7%)	0.6641
Yes	22 (23.9%)	23 (26.7%)	45 (25.3%)	
Mean (SD)	27.5 (6.4)	27.4 (5.6)	27.4 (6.0)	0.9681
Median (IQR)	25.5 (23.5, 29.8)	26.0 (23.4, 30.7)	25.8 (23.5, 30.0)	0.781
**Racial background, n (%)**				
Caucasian	81 (88.0%)	79 (91.9%)	160 (89.9%)	0.3986
Other	11 (12.0%)	7 (8.1%)	18 (10.1%)	
**Smoking status, n (%)**				
Never	56 (60.9%)	47 (54.7%)	103 (57.9%)	0.6023
Previous	33 (35.9%)	37 (43.0%)	70 (39.3%)	
Current	3 (3.3%)	2 (2.3%)	5 (2.8%)	
**Comorbidities, diabetes or cardiovascular or respiratory or cancer n (%)**				
No	47 (51.1%)	36 (41.9%)	83 (46.6%)	0.2175
Yes	45 (48.9%)	50 (58.1%)	95 (53.4%)	
**Vaccine doses prior to baseline, n (%)**				
2	2 (2.2%)	0 (0.0%)	2 (1.1%)	0.0038
3	13 (14.1%)	10 (11.6%)	23 (12.9%)	
4	42 (45.7%)	20 (23.3%)	62 (34.8%)	
5	35 (38.0%)	55 (64.0%)	90 (50.6%)	
6	0 (0.0%)	1 (1.2%)	1 (0.6%)	
Mean (SD)	4.20 (0.76)	4.55 (0.71)	4.37 (0.76)	0.0018
Median (IQR)	4.00 (4.00, 5.00)	5.00 (4.00, 5.00)	5.00 (4.00, 5.00)	0.0009
**Vaccination within 4 month follow up, n (%)**				
No	68 (73.9%)	60 (69.8%)	128 (71.9%)	0.5386
Yes	24 (26.1%)	26 (30.2%)	50 (28.1%)	
**Breakthrough COVID-19 infection**				
Number of breakthrough infections before baseline				
Mean (SD)	1.04 (0.21)	-	0.54 (0.54)	-
Median (IQR)	1.00 (1.00, 1.00)	-	1.00 (0.00, 1.00)	-
Time from last breakthrough or last vaccine to substudy baseline				
Mean (SD)	156.7 (98.1)	105.8 (97.4)	132.1 (100.8)	0.0007
Median (IQR)	145 (70.5, 236.5)	76.5 (40, 133)	93 (53, 197)	0.0002
Breakthroughs between baseline and 4 month follow up, n (%)				
No	84 (91.3%)	62 (72.1%)	146 (82.0%)	0.0009
Yes	8 (8.7%)	24 (27.9%)	32 (18.0%)	

**Table 2 vaccines-12-00564-t002:** The proportion of participants with neutralizing antibody detection and levels at baseline for Wuhan and Omicron variants BA.5. XBB.1.5, and EG.5 with hybrid/vaccine-only immunity at baseline, SD—standard deviation, IQR—interquartile range, n—number.

	Hybrid Immunity	Vaccine Only Immunity	Total	*p*-Value
**Cohort, N**	**92**	**86**	**178**	**-**
**Wuhan: Antibody, n (%)**				
Non-detectable	-	-	-	-
Detectable	92 (100%)	86 (100%)	178 (100%)	-
Log 10 (1/ID50) for detectables				
Mean (SD)	4.28 (0.95)	3.79 (0.71)	4.05 (0.87)	0.0001
Median (IQR)	4.45 (3.90, 4.78)	3.79 (3.37, 4.23)	4.02 (3.58, 4.54)	<0.0001
**Omicron BA.5: Antibody, n (%)**				
Non-detectable	2 (2.2%)	12 (14.0%)	14 (7.9%)	0.0035
Detectable	90 (97.8%)	74 (86.0%)	164 (92.1%)	
Log 10 (1/ID50) for detectables				
Mean (SD)	3.64 (0.54)	2.62 (0.58)	3.18 (0.76)	<0.0001
Median (IQR)	3.70 (3.29, 4.00)	2.59 (2.18, 2.96)	3.19 (2.61, 3.78)	<0.0001
**Omicron XBB.1.5: Antibody, n (%)**				
Non-detectable	10 (10.9%)	52 (60.5%)	62 (34.8%)	<0.0001
Detectable	82 (89.1%)	34 (39.5%)	116 (65.2%)	
Log 10 (1/ID50) for detectables				
Mean (SD)	2.80 (0.47)	2.10 (0.50)	2.59 (0.57)	<0.0001
Median (IQR)	2.80 (2.51, 3.15)	1.91 (1.78, 2.41)	2.63 (2.09, 3.03)	<0.0001
**Omicron EG.5: Antibody, n (%)**				
Non-detectable	16 (17.4%)	61 (70.9%)	77 (43.3%)	<0.0001
Detectable	76 (82.6%)	25 (29.1%)	101 (56.7%)	
Log 10 (1/ID50) for detectables				
Mean (SD)	2.62 (0.49)	2.04 (0.41)	2.48 (0.53)	<0.0001
Median (IQR)	2.62 (2.25, 2.98)	1.91 (1.79, 2.20)	2.50 (2.05, 2.92)	<0.0001

**Table 3 vaccines-12-00564-t003:** Predictors of increases in neutralizing antibody titer during 4-month follow up. OR—odds ratio, adOR—adjusted odds ratio, FU—follow up.

COVID-19 Strain	Wuhan	Omicron BA.5	Omicron XBB.1.5	Omicron EG.5
Age: 70+ years				
Univariable (OR)	0.55 (0.29, 1.06)	0.71 (0.39, 1.32)	0.71 (0.39, 1.32)	1.04 (0.58, 1.89)
Multivariable (adOR)	0.42 (0.18, 0.99)	0.61 (0.28, 1.29)	1.24 (0.6, 2.58)	1.01 (0.49, 2.11)
Gender: Male				
Univariable (OR)	1.27 (0.65, 2.47)	0.9 (0.47, 1.73)	0.9 (0.47, 1.73)	0.77 (0.42, 1.42)
Multivariable (adOR)	1.22 (0.56, 2.65)	0.91 (0.45, 1.85)	0.55 (0.27, 1.09)	0.59 (0.29, 1.17)
BMI: Obese				
Univariable (OR)	0.36 (0.17, 0.77)	0.88 (0.46, 1.66)	0.88 (0.46, 1.66)	0.92 (0.5, 1.69)
Multivariable (adOR)	0.27 (0.11, 0.65)	0.89 (0.45, 1.77)	0.94 (0.48, 1.81)	0.91 (0.46, 1.75)
Comorbidity: Yes				
Univariable (OR)	0.66 (0.36, 1.2)	0.85 (0.48, 1.49)	0.85 (0.48, 1.49)	1.1 (0.65, 1.88)
Multivariable (adOR)	1 (0.46, 2.17)	1.1 (0.55, 2.19)	1.02 (0.53, 1.98)	1.43 (0.74, 2.8)
Arm: Hybrid immunity				
Univariable (OR)	1.35 (0.74, 2.46)	0.75 (0.42, 1.33)	0.75 (0.42, 1.33)	1.44 (0.83, 2.54)
Multivariable (adOR)	1.14 (0.57, 2.28)	1.5 (0.79, 2.9)	3.39 (1.78, 6.64)	3.73 (1.95, 7.34)
4M FU period vaccination				
Univariable (OR)	4.12 (2.06, 8.24)	3.81 (1.98, 7.49)	3.81 (1.98, 7.49)	2.12 (1.16, 3.88)
Multivariable (adOR)	5.79 (2.64, 12.72)	7.08 (3.39, 15.28)	5.58 (2.78, 11.58)	4.43 (2.26, 8.91)
4M FU period breakthrough COVID-19 infection				
Univariable (OR)	2.07 (0.95, 4.48)	10.01 (4.4, 24.68)	10.01 (4.4, 24.68)	22.94 (9.86, 57.48)
Multivariable (adOR)	2.85 (1.16, 7)	20.11 (7.94, 55.33)	77.94 (28.04, 236.54)	73.96 (26.66, 224.96)

**Table 4 vaccines-12-00564-t004:** Decay slope in neutralizing antibody levels between baseline and 4 months to the Wuhan and Omicron variants in participants who did not receive an additional vaccine or experience a breakthrough infection between sampling times based on the presence of vaccine only or hybrid immunity.

COVID-19 Strain	Hybrid-Immunity Decay Slope of Neutralizing Antibody Levels (*p* Value)	Vaccine-only Immunity Decay Slope of Neutralizing Antibodies (*p* Value)
Wuhan	−0.0928 (0.53)	−0.5006 (0.001)
Omicron BA.5	−0.2033 (0.10)	−0.8799 (0.001)
Omicron XBB.1.5	−0.3143 (0.06)	−0.6021 (0.011)
Omicron EG.5	−0.2023 (0.29)	−0.4050 (0.044)

**Table 5 vaccines-12-00564-t005:** Predictors for protection against breakthrough COVID-19 infection in short (4-month)/long term (10-month) follow up. HR—hazard ratio, adHR—adjusted hazard ratio.

Predictors of Breakthrough COVID-19 Infection	Univariable (HR)	Multivariable (adHR)	
		Including Hybrid Status	Including BA.5 Neutralizing Antibody Level
Short term (4 months)			
Age: 70+	1.14 (0.53, 2.47)	1.11 (0.46, 2.69)	1.22 (0.5, 2.97)
Gender: Male	1.1 (0.51, 2.38)	1.16 (0.52, 2.63)	1.09 (0.48, 2.44)
BMI: Obese	0.69 (0.28, 1.67)	0.66 (0.27, 1.62)	0.85 (0.34, 2.1)
Comorbidity: Yes	1 (0.5, 2)	0.72 (0.32, 1.58)	0.78 (0.35, 1.74)
Arm: Hybrid	0.27 (0.12, 0.61)	0.24 (0.11, 0.54)	-
BA.5 antibody level at baseline	0.76 (0.59, 0.99)	-	0.72 (0.55, 0.94)
4M FU period vaccination	0.41 (0.16, 1.08)	0.36 (0.14, 0.94)	0.35 (0.13, 0.93)
Long term (10 months, up to 22 September 2023)			
Age: 70+	1.07 (0.63, 1.83)	1.13 (0.6, 2.11)	1.14 (0.61, 2.15)
Gender: Male	0.9 (0.52, 1.57)	0.96 (0.54, 1.72)	0.95 (0.53, 1.69)
BMI: Obese	0.98 (0.56, 1.7)	1.08 (0.62, 1.89)	1.12 (0.63, 1.97)
Comorbidity: Yes	1.3 (0.8, 2.11)	1.24 (0.71, 2.18)	1.31 (0.75, 2.29)
Arm: Hybrid	0.64 (0.4, 1.04)	0.59 (0.36, 0.96)	-
BA.5 antibody level at baseline	0.97 (0.78, 1.19)	-	0.92 (0.74, 1.14)
Vaccination dose to 22 September 2023	0.43 (0.25, 0.74)	0.38 (0.22, 0.67)	0.4 (0.23, 0.7)

## Data Availability

The relevant anonymized data have been transferred to the Canadian COVID-19 Immunity Task Force (CITF) as part of a standard data-sharing agreement with the Public Health Agency of Canada. The dataset has been submitted together with a dictionary defining each variable in the dataset. CITF will store the dataset in its Database which will be held under the custodianship of McGill University to be shared nationally and internationally. Access to all CITF data for external researchers will be available by submitting a request to the data access committee of CITF. CITF will evaluate the request by a checklist to ensure that the request follows the privacy and ethical protocols and is in compliance with Canadian law and research ethics. Data in the CITF Database can be used by researchers across Canada and other countries following Data Access Committee (DAC) approval. The dataset may also be shared with other research databases following similar procedures and protections.

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
