# Peer review of "Declining Levels of Neutralizing Antibodies to SARS-CoV-2 Omicron Variants Are Enhanced by Hybrid Immunity and Original/Omicron Bivalent Vaccination"

_vaccines, 2024, doi:10.3390/vaccines12060564_

Round 1

Reviewer 1 Report

Comments and Suggestions for Authors

The paper describes the sub-study conducted on the cohort of adults vaccinated against SARS-CoV-2. It shows that although all the participants had detectable neutralizing antibodies against Wuhan SARS-CoV-2, the number of adults with detectable Nabs against later viral variants declined, as did the levels of Nabs. In addition, those with hybrid immunity had higher levels of Nabs than those with vaccine-only immunity, but bivalent vaccine provided higher levels of Nabs to omicron variants. Finally, both bivalent vaccine-induced and hybrid immunity were associated with protection during a 4-10 months of follow-up.

The study is scientifically sound, well done and the limitations are stated, but there are some comments.

1. It would be easier to understand the methodology of the study if the authors presented its timeline in a schematic form (vaccination, breakthrough infections, repeated vaccination, sample collection, etc.)

2. Why was the 50-70 age group not included in the study?

3. How would you explain the phenomenon that Nabs responses were not affected by age and comorbidity?

 4. There are some typos and word omissions in the text (e.g., line 211).

Author Response

Please see the attached pdf file.

Reviewer 2 Report

Comments and Suggestions for Authors

Walmsley an Co-workers present 4 major findings: In heavily vaccinated individuals, neutralizing antibodies are considerably lower to Omicron variants than to the original strain of SARS-CoV2 ("Wuhan").    Hybrid immunity is associated with higher levels of neutralizing antibodies against all variants than immunity induced by vaccination alone.  Increase of neutralizing antibody titers was more pronounced after breakthrough infections than after an additional vaccination. The level of antibodies to BA5 was not predictive of breakthrough infection (although this finding is not clear).

The latter finding would be new as several previous studies indeed find that the level of neutralizing antibodies is predictive of breakthrough infections (e.g. Seelischer et al, Lancet Microbe 2023).

The other findings are all not new and published already in several previous studies. The presented data are thus mainly confirmatory.

Major points

1. The neutralizing antibody assay should be validated with a standard. Result should be presented as suppl. data (eg Feng Zhu, Lancet Microbe, 2022).

2. The individuals with breakthrough infections and additional vaccination should not be group with the rest of the study participants in figure 4.

3. The finding that neutralizing antibodies to BA5 were not predictive for breakthrough infection seems contradictory to the claim in the last paragraph that higher neutralizing antibodies were associated with lower breakthrough infections. Is this because hybrid immunity is protective, which is associated with higher levels of neutralizing antibodies, but antibodies are not an independent predictor? This should be clarified. 

4. The authors should more clearly point out in the discussion which data are actually new. 

Author Response

Please see the attached pdf file

Round 2

Reviewer 1 Report

Comments and Suggestions for Authors

The authors have responded to the reviewer's comments and added the suggested scheme to the text. The article can be accepted as it is now. 

Reviewer 2 Report

Comments and Suggestions for Authors

The authors have now improved the manuscript. Although the results are mostly confirmatory, there is some value in publishing this work.